# The Topical Novel Formulations of Interferon α-2в Effectively Inhibit HSV-1 Keratitis in the Rabbit Eye Model and HSV-2 Genital Herpes in Mice

**DOI:** 10.3390/v16060989

**Published:** 2024-06-19

**Authors:** Anna Ivanina, Irina Leneva, Irina Falynskova, Ekaterina Glubokova, Nadezhda Kartashova, Nadezda Pankova, Sergei Korovkin, Oxana Svitich

**Affiliations:** 1Mechnikov Research Institute of Vaccines and Sera, Department of Virology, 105064 Moscow, Russia; wnyfd385@yandex.ru (I.L.); falynskova@mail.ru (I.F.); eaglubokova@yandex.ru (E.G.); nadezdakartasova10571@gmail.com (N.K.); svitichoa@yandex.ru (O.S.); 2OOO Firn M, Biotech Company, 108804 Moscow, Russia; nadezhda.pankova.nnov@gmail.com (N.P.); korovkin09@mail.ru (S.K.)

**Keywords:** HSV-1, HSV-2, interferon alpha-2b, topical formulations, herpetic stromal keratitis, genital herpes infection

## Abstract

Herpes simplex viruses type 1 (HSV-1) and type 2 (HSV-2) are widespread human pathogens that establish chronic latent infections leading to recurrent episodes. Current treatments are limited, necessitating the development of novel antiviral strategies. This study aimed to assess the antiviral efficacy of novel topical formulations containing interferon alpha-2b (IFN α-2b) against HSV-1 and HSV-2. The formulations, Oftalmoferon^®^ forte (eye drops) and Interferon Vaginal Tablets, demonstrated potent antiviral effects against HSV-1 and HSV-2 in Vero cells, respectively, with concentration-dependent inhibition of viral replication. Subsequently, their efficacy was tested in animal models: HSV-1 keratitis in the rabbit eye model and HSV-2 genital herpes in mice. Oftalmoferon^®^ forte effectively treated HSV-1 keratitis, reducing clinical symptoms and ulcerations compared to virus control. Interferon Vaginal Tablets showed promising results in controlling HSV-2 genital herpes in mice, improving survival rates, reducing clinical signs, weight loss and viral replication. The novel IFN α-2b formulations exhibited significant antiviral activity against HSV infections in cell culture and animal models. These findings suggest the potential of these formulations as alternative treatments for HSV infections, particularly in cases resistant to current therapies. Further studies are warranted to optimize treatment regimens and assess clinical efficacy in humans.

## 1. Introduction

Herpes simplex virus type 1 (HSV-1) and type 2 (HSV-2) are highly prevalent human pathogens with global prevalence rates of about 67% and 13%, respectively [1]. The main danger of herpes infection lies in the fact that, after the initial infection, the disease passes into a chronic latent stage, and the virus persists in the patient’s body throughout life, relapsing every time the human immune system weakens [2,3,4,5,6].

HSV-1 and HSV-2 are causative agents of various diseases, prominently including herpetic stromal keratitis and genital herpes. Herpetic stromal keratitis poses a significant threat, with approximately one in five individuals with ocular HSV infection progressing to this vision-threatening condition [7]. In fact, herpetic keratitis is considered to be the leading cause of monocular infectious blindness due to stromal opacification [8]. Beyond the ominous threat of blindness, HSV keratitis can manifest in various debilitating ways, including dry eye disease, persistent pain, corneal surface irregularities, ulcerations and, in rare instances, corneal perforation [9]. Genital herpes, caused by HSV-2, is one of the most prevalent sexually transmitted infections that disproportionately impacts women worldwide [10,11,12,13]. Genital HSV-2 infection initiates with lytic replication in vaginal mucosal keratinocytes, resulting in the development of genital lesions ranging from macules and papules to ulcers, pustules and vesicles [14,15,16]. Extended infections can cause flu-like symptoms and eventually lymphadenopathy, cervicitis and proctitis [2,17].

Despite the widespread incidence of HSV infections, no definitive preventive or curative treatments have been developed. Current therapeutic strategies primarily rely on antiviral agents for reactivation suppression; however, herpetic stromal keratitis and genital herpes persist as lifelong infections. Moreover, the emergence of resistance to antivirals such as Acyclovir, particularly in immunocompromised individuals, poses a significant challenge [18].

Interferons (IFNs) play a pivotal role as a primary immune defense mechanism against viral infections, crucial for controlling viral replication and dissemination, especially at mucosal surfaces that serve as primary sites of pathogen exposure [19,20]. The inhibition of viral propagation and spread at mucosal sites is essential for preventing severe disease outcomes while mitigating potential inflammatory repercussions. Topical formulations of recombinant IFN α-2b have been licensed and are widely used in Russia for treating HSV infections, including ocular keratitis and urogenital diseases [21,22]. The advantages of topical formulations include patient convenience, localized high drug concentrations and minimized systemic toxicity. Topical antiviral formulations are particularly desirable due to improved patient compliance with simplified dosing regimens or lower drug concentrations. Hence, there is a critical need for the development of new topical antiviral formulations that would represent a significant improvement over the currently available antiviral drugs.

Novel formulations of IFN, namely Oftalmoferon^®^ forte eye drops for HSV-1 keratitis and Interferon Vaginal Tablets for HSV-2 genital herpes, have been developed by OOO Firn M.

Oftalmoferon^®^ forte combines IFN and azelastine to inhibit viral replication and allergic reactions, respectively. Azelastine prevents the development of allergic reactions by inhibiting the histamine H1 receptor, preventing the release of histamine and the activation of other inflammatory mediators [23,24,25,26,27]. Interferon Vaginal Tablets incorporate IFN along with sodium bicarbonate and succinic acid to enhance antiviral properties through the generation of foam upon contact with vaginal moisture, facilitating rapid IFN dispersion over the vaginal mucosa for heightened therapeutic efficacy [28].

In this study, we present the antiviral activity of novel IFN α-2b topical formulations against HSV-1 and HSV-2 in cell culture, along with their efficacy in animal models of HSV-1 keratitis in the rabbit eye model and HSV-2 genital herpes in mice.

## 2. Materials and Methods

### 2.1. Viruses and Cells

Herpes simplex virus type 1 (HSV-1) strain VR-3 and HSV-2 strain VN obtained from the State Collection of Viruses of the D.I. Ivanovsky Research Institute of Virology was used. Vero cells (Green Monkey Kidney cells) obtained from the ATCC (American Collection of Cell Cultures and Viruses) and grown at 37 °C, with 5% CO_2_, in Dulbecco’s Modified Eagle Medium (DMEM, Pan Eco, Moscow, Russia) and supplemented with 10% heat-inactivated fetal bovine serum (Pan Eco, Moscow, Russia), 2 mmol/L L-glutamine (Pan Eco, Moscow, Russia), 100 U/mL penicillin and 100 µg/mL streptomycin. Viral stocks with titers 10^2^TCID_50_/mL, 10^3^TCID_50_/mL and 10^6^TCID_50_/mL were used for antiviral activity studies and animal experiments, respectively.

### 2.2. Antivirals

Oftalmoferon^®^ forte (OOO Firn M): A combined drug comprising human recombinant IFN alpha-2b (≥500,000 IU/mL) along with azelastine (0.25 mg/mL); Interferon Vaginal Tablets for intravaginal administration, containing human recombinant IFN α-2b at a dosage of 500,000 IU/tablet; Acyclovir (ACV) in the form of a 3% eye ointment (Sintez, Kurgan, Russia) and a 5% cream (Vertex, Saint-Petersburg, Russia).

### 2.3. Determination of Cytotoxicity in Vero Cells

Cytotoxicity assessment of the formulations was conducted using the MTT assay. Briefly, Vero cells at a density of 2 × 10^4^ cells/well were seeded into flat-bottom, 96-well microtiter plates and incubated until formation of a confluent monolayer (37 °C, 5% CO_2_). A range of concentrations of Interferon alfa 2-b (0.001 to 10,000 IU/mL) was prepared using cell culture medium and added to plates in quadruplicate (200 μL). After 72 h, the treatments were removed, and 40 μL of MTT reagent (5 µg/mL) was added to each well and incubated for a further 2 h. Media were then removed, and 100 μL of DMSO solution was added to the wells. Finally, plates were read at 550 nm by a microplate reader (Varioskan Flash, Thermo Scientific, Waltham, MA, USA). The percentage cell viability was calculated using the following formula: Cell viability (%) = [(OD of untreated cells − treated cells)/(OD of untreated cells)] × 100. The 50% cytotoxicity concentration (CC_50_) was defined as follows: the cytotoxic concentration of each compound that reduced the absorbance of treated cells to 50% when compared with that of untreated cells.

### 2.4. Determination of Antiviral Activity against HSV-1 and HSV-2 in Vero Cells

Antiviral activity against HSV-1 and HSV-2 was determined using cytopathic effect (CPE) inhibition assay. Confluent monolayers of Vero cells (2 × 10^4^ cells/well) in flat-bottom, 96-well microtiter plates were pre-incubated with 100 μL of maintenance medium, without and with different non-toxic drug concentrations in quadruplicate for 2 “h” at 37 °C. The “maintenance medium” featured the same composition as the “growth medium”, except for the concentration of FCS (5%). “Virus controls” (infected, but untreated) and “cell controls” (uninfected, untreated cells) were included on each plate prepared throughout the experiment. After 2 “h” of incubation, HSV-1 and HSV-2 (10 and 10^2^TCID_50_, in 100 μL, respectively) were added to wells, except “cell control” wells, and cells were incubated at 37 °C, 5% CO_2_ with humidification. Cells were evaluated once daily by light microscope. After development of complete CPE in viral control wells (72 “h”), the medium was removed, and 40 μL of MTT reagent (5 µg/mL) was added to each well and incubated for a further 2 “h”. Then, the inoculum was discarded and 100 μL of DMSO solution was added to the wells. Finally, the plates were read at 530 and 620 nm by a microplate reader (Varioskan Flash, Thermo Scientific, Waltham, MA, USA). The EC_50_ was defined as follows: the inhibitory concentration of compound that reduced the absorbance of treated infected cells to 50%, when compared with that of cell controls.

### 2.5. Ethical Considerations

All animal care and procedures were performed in accordance with institutional and Russian Ministry of Health guidelines and approved by the Animal Care Committee at the I.I Mechnikov Research Institute for Vaccines and Sera, protocol №11, 28 July 2023.

### 2.6. HSV-1 Keratitis in the Rabbit Eye Model

Female rabbits of the Soviet chinchilla, 2.5–3 months old (2 kg) were obtained from “STEZAR” (Vladimir Region). A day before infection, all rabbits received topical eye treatment three times per day with 0.1% tetracycline to reduce bacterial contamination, and 40 s before infection, a 0.5% alkaline solution was applied to the cornea of the eye for anesthesia. The scarified (2 × 2 cross-hatch pattern) corneas of rabbits were injected into each eye with 50 μL of a suspension of HSV-1 strain with viral titer 10^6^TCID_50_/mL. The eyelids were closed and the viral suspension rubbed gently on the corneal surface for 40 s. Care was taken to avoid leakage of the suspension. Three groups of 10 rabbits each were used, one group each for treatment with Oftalmoferon^®^ forte, Acyclovir (ACV, eye ointment of 3%) and a vehicle control (PBS). Both eyes of all rabbits were used. Oftalmoferon^®^ forte and ACV (2 caplets 50 μL at concentration of 1%) were applied topically six and five times per day, respectively. PBS was applied as an Oftalmoferon^®^ forte. Treatment began on postinfection day 2 and continued for five consecutive days. Slit lamp examination (SLE) was performed on a masked basis once a day with pre-instillation of fluorescein to visualize lesions. The rabbits were monitored for clinical signs using a scoring system described in the Results Section.

### 2.7. Mouse Model of HSV-2 Vaginal Herpes Infection

The 6 weeks-old (12–14 g) female BALB/c mice obtained from “STEZAR” (Vladimir Region) were randomized into 6 groups, and each group contained 10 mice. There were four groups treated with Interferon Vaginal Tablets containing next dosages of IFN alpha 2-b: 6,250,000 IU/kg (1/4 tablets), 4,166,667 IU/kg (1/6 tablets), 3,125,000 IU/kg (1/8 tablets), 2,500,000 IU/kg (1/10 tablets), viral control group treated with PBS and a positive control group treated with Acyclovir (ACV, cream of 5%). The study dosages of vaginal tablets were administered intravaginally, using a total volume of 0.2 mL per mouse of HSV-2 stock (virus titer 10^6^TCID_50_/mL). The vaginal epithelium was scarified after local anesthesia with lidocaine before infection. Treatment started after 4 h post infection, and the drugs were administered 1 time every day for 9 days. Vaginal swabs were collected at 5 and 10 days post-infection from 3 mice from each group to assess the viral titer in the vaginal epithelium. Animals were monitored for any change in behavior, clinical signs, death and weight loss for 21 days. Clinical signs were assigned using the scoring system described in the Results Section.

### 2.8. Statistical Analysis

Statistical analysis was conducted using GraphPad Prism version 5.03 software. The data were expressed as the mean ± standard deviation (SD). Normality of the data distribution was assessed using the Shapiro–Wilk test. To compare the statistical significance of differences between experimental and control groups, the Kruskal–Wallis test with Bonferroni correction was employed. Kaplan–Meier curves were used for visual representation of survival data. Differences were considered significant if *p* < 0.05.

## 3. Results

### 3.1. Cytotoxic Effect and Antiviral Activity of IFN Alpha 2-b Formulations in Vero Cells

A cytotoxicity assay was conducted to determine the non-toxic concentrations of novel formulations of INF alfa 2-b in Vero cells used in our research. The CC_50_ values for Oftalmoferon^®^ forte and Interferon Vaginal Tablets were found to be 56,647.089 IU/mL and 5437.53 ± 530.74 IU/mL, respectively (Figure 1, Table 1). Thus, the cytotoxicity of Interferon Vaginal Tablets was higher than the cytotoxicity of Oftalmoferon^®^ forte, but the CC_50_s of the studied formulations were in the range of values previously described for IFN substances [29,30].

The impact of these formulations on viral replication in Vero cells was assessed using two distinct strains of HSV, which were subsequently employed to evaluate efficacy in animal models: Oftalmoferon^®^ forte against HSV-1 and Interferon Vaginal Tablets against HSV-2. Both Oftalmoferon^®^ forte and Interferon Vaginal Tablets exhibited inhibitory effects on the replication of HSV-1 and HSV-2, respectively. Notably, the antiviral effects against both HSV strains displayed a concentration-dependent relationship, strengthening with increasing drug concentrations (Figure 1). The EC_50_ and EC_90_ values derived from dose–response curves were determined to be 2.60 IU/mL and 38.16 ± 1.58 IU/mL for Oftalmoferon^®^ forte against HSV-1 and 6.98 ± 2.38 UI/mL and 62.82 ± 1.80 UI/mL for Interferon Vaginal Tablets against HSV-2, respectively (Figure 1, Table 1). Despite the fact that, due to differences in cytotoxicity, the selectivity index (SI) values differed significantly, they were high enough for both drugs. The SIs for Oftalmoferon^®^ forte and Interferon Vaginal Tablets were 21741 and 788, respectively.

### 3.2. Efficacy of Topical Treatment Oftalmoferon^®^ Forte against HSV-1 Keratitis in the Rabbit Eye Model

Topical treatment with Oftalmoferon^®^ forte drops and daily assessment of the rabbit eyes were started on day 2 post-infection (Figure 2A), after the appearance of the first pathological changes, which were scored with ophthalmoscopy and compared with controls, as described in Table 2 and Table 3.

The visible difference between the manifestations of herpes infection in the virus control group and the group treated with Oftalmoferon^®^ forte became apparent on the fourth day of therapy. Notably, in the treated group, the symptoms began to decrease significantly. The outflow from the eyes changed from purulent–fibrinous to catarrhal, and in the virus control group the number of purulent outflows continued to increase daily. By the seventh day after beginning therapy, the clinical symptoms of infection in the group with therapy disappeared, while in the virus control group there were signs of iritis, uveitis, mixed vascularization, corneal edema and erosive lesions (Figure 2B–D). Importantly, on day 9 after infection, one rabbit died with signs of nervous system disorders (abundant salivation, involuntary twitching of the head and limbs, tachycardia, stupor) in the virus control group, whereas no deaths were recorded in the treated group.

The severity of HSV-1 infection in eyes treated with Oftalmoferon^®^ forte was notably lower by 13% and 58% (*p* ≤ 0.05) on days 4 and 7 post-treatment initiation, respectively, compared to untreated eyes (Figure 2E). On the contrary, ACV was more effective than Oftalmoferon^®^ forte in reducing the severity of infection, and its effect was observed immediately after the start of therapy (Figure 2E).

Furthermore, we used ulceration of the cornea as a separate criterion, because these lesions reflected stromal involvement in keratitis induced by HSV-1 (Table 2 and Table 3). The assessment based on corneal ulceration revealed a significant reduction in ulcer severity in Oftalmoferon^®^ forte-treated eyes from day 3 onwards (*p* < 0.05), with a substantial difference of 93% observed on day 7 (*p* < 0.05) (Figure 2). By the end of the study, ulcerations occupied up to one-third of the cornea in the virus control group, while two out of three rabbits treated with Oftalmoferon^®^ forte exhibited complete corneal regeneration, with minor ulcers healing within days.

Although ACV initially demonstrated superior efficacy in the early stages of treatment, from the fourth day onwards, the severity of ulcerations in Oftalmoferon^®^ forte-treated eyes were approximately equal to those in ACV-treated eyes (Figure 2F).

In conclusion, our rabbit ocular study using topical treatment with Oftalmoferon^®^ forte demonstrated significant therapeutic efficacy against HSV-1-induced keratitis in rabbits. This treatment fully protected animals against death and significantly reduced the severity of infection compared to the placebo group. Oftalmoferon^®^ forte was as effective as ACV in reducing the severity of ulcerations induced by HSV-1.

### 3.3. Efficacy of Interferon Vaginal Tablets in Mice with Genital Infection Induced by HSV-2

Following the confirmation of the antiviral efficacy of Interferon Vaginal Tablets in vitro, our study aimed to evaluate the therapeutic potential of topical Interferon Vaginal Tablets in protecting mice from a murine model of HSV-2-induced genital infection (Figure 3A). Daily monitoring of the animals was conducted, and the antiviral effects were assessed over a 20-day period post-infection using a scoring system summarized in Table 4.

In the virus control group, clinical symptoms of genital infection began to manifest on the second day post-infection, characterized by notable swelling of the genital fissure, tissue hyperemia and the presence of flaky whitish discharge. In the viral control group, by the fourth day post-infection, signs of damage to the peripheral nervous system were observed, leading to motor deficits and neurological impairments. Starting from the eighth day after infection, lesions of the central nervous system of the spinal nerves led to the death of animals with characteristic clinical signs. Clinical signs in the virus control group were estimated between 2.6 and 19.6 points from day 2 to day 20 (Figure 3B). By the end of the experiment, a high mortality rate (80%) and loss of body weight (16%) were observed in the virus control group (Figure 3C,D).

The clinical signs of vaginal herpes in all treated groups estimated in scores were less pronounced than in the virus control with varying degrees of improvement based on the administered dosage. The smallest difference with viral control was observed in the group treated at the lowest dose of 2,500,000 IU/kg. In groups treated with Interferon Vaginal Tablets at high doses, the difference with the virus control group was higher. By 6–8 days after infection, the difference between the therapy groups and the viral control group averaged 6–10 scores, and by day 20 the difference grew (*p* ≤ 0.05). Treatment with the low dose significantly enhanced survival (20%, *p* < 0.05) and administration in other doses fully protected mice from death and symptoms of nervous phenomena. Additionally, treatment with Interferon Vaginal Tablets at all studied doses abolished the weight loss (Figure 3C). The efficacy of Interferon Vaginal Tablets at doses of 6 250,000 IU/kg, 4,166,667 IU/kg and 3,125,000 IU/kg, estimated by clinical signs, survival rate and loss weight was similar to the efficacy of Acyclovir taken as a control drug (Figure 3C). However, the complete disappearance of clinical symptoms in the group treated with ACV occurred earlier (on day 12 after infection compared with on day 16–18 after infection in groups treated with Interferon Vaginal Tablets (Figure 3B)).

To determine effect of Interferon Vaginal Tablets on the presence of productive virus at the primary site of infection and local shedding of infectious virions, vaginal swabs were taken 5 and 10 days post-infection (dpi) from all of the groups to assess the extent of viral spread through a CPE assay. Viral shedding analysis of vaginal swabs revealed significantly lower viral titers in mice treated with Interferon Vaginal Tablets at higher doses (6,250,000 IU/kg, 4,166,667 IU/kg and 3,125,000 IU/kg) and ACV compared to the virus control group. In the group treated with a dose of 2,500,000 IU/kg, the virus titer was higher than in other treated groups, but lower than in the virus control group (2.9 lgTCID_50_/0.1 mL vs. 4.2 lgTCID_50_/0.1 mL), although this difference was not statistically significant (Figure 3E).

The reduction in viral replication correlated with improved survival rates, reduced weight loss and an alleviation of clinical symptoms. These findings underscore the potent antiviral activity of Interferon Vaginal Tablets in suppressing HSV-2 genital infection in mice, highlighting their potential as an effective intravaginal treatment option against HSV-2.

## 4. Discussion

Herpes simplex virus type 1 (HSV-1) and type 2 (HSV-2) belong to the neurotropic alphaherpesvirus subfamily of herpesviruses. They share strong genetic homology, and both viruses result in very similar innate and adaptive immune responses from the human hosts. While ACV and related nucleoside analogues provide successful modalities for treatment and suppression, HSV remains highly prevalent worldwide. The emergence of ACV-resistant virus strains and the universal ability of HSV to establish latency coupled with adverse effects of long-term systemic use of currently available antiherpetic compounds provide a stimulus for an increased search for new and more effective antivirals against HSV. Interferon (IFN) exhibits pleiotropic antiviral effects by directly inhibiting viral replication and stimulating innate and adaptive immune responses [19,20]. IFN level is not always enough to suppress the infection, especially in immunosuppressed hosts. Therefore, the aim of the treatment with IFN isto enhance the action of endogenous cytokines by introducing exogenic IFN. However, the polypeptide structure of IFN limits its oral administration, and systemic delivery poses challenges due to its short half-life. To overcome these limitations, novel formulations of IFN with topical application have been developed to enhance efficacy and minimize side effects.

Here, we report the antiviral activity of novel topical formulations of IFN α-2в against HSV-1 and HSV-2 in cell culture and efficacy in animal models: against HSV-1 keratitis in the rabbit eye model and HSV-2 genital herpes in mice.

At first, we studied the antiviral activity of novel formulations containing IFN α-2в against HSVs that we used in animal models. Our initial results from this study show that Oftalmoferon^®^ forte and Interferon Vaginal Tablets effectively inhibited the replication of HSV-1 and HSV-2 in Vero cells, respectively. Another interesting result that we observed in this study was the tolerability of both formulations in cells. In this study, we observed that, even at a concentration as high as 5000 IU/mL, both drugs did not affect the viability of cells, giving us the confidence that they can have a large therapeutic window for the treatment of HSV infections in vivo.

Subsequent animal studies revealed the efficacy of these formulations in treating HSV-1 keratitis in a rabbit eye model and HSV-2 genital herpes in mice. The efficacy of the new topical formulation of IFN α-2в Oftalmoferon^®^ forte was evaluated against HSV-1 keratitis in the rabbit eye model. It should be noted that HSV-1 corneal infections could be studied in a variety of animal models of which the most popular has been developed in rabbits. This model has a large eye and spontaneous HSV-1 reactivation from latency and shedding at the ocular surface. In addition, infectious epithelial keratitis typically persists longer in the rabbit, which can be helpful in testing the efficacy of antiherpetic drugs. The assessment of Oftalmoferon^®^ forte in the rabbit eye model of HSV-1 keratitis showed promising results, with a significant reduction in clinical symptoms and ulcerations compared to the virus control group.

In our study, we used a modified scoring estimation, which was different from the system used before [31]. In addition to using the system to estimate all pathologic changes, we used the criterion of the severity of ulcerations of cornea, since this sign is one of the main manifestations of ophthalmic herpes [32]. This is due to the fact that HSV-1 persists and spreads along nerve fibers [33], and the cornea of the eye is one of those tissues in which the number of nerve fibers is especially high. HSV-1 actively multiplies in the corneal epithelium, resulting in cytopathic and dystrophic processes, during which epithelial cells necrotize and peel off, forming ulcerations [34]. Taking into consideration the specificity of the development of herpetic keratitis in rabbits, namely the predominance of diffuse ulcerations under tree-like lesions, the criteria for evaluating these lesions were changed in the scoring system. In addition, we observed the death of a rabbit in the viral control group. Presumably, the virus from the cornea of the eye entered the ganglia of the trigeminal nerve, and from there into the brain, causing the inflammation and death of the animal with pronounced nervous phenomena [35,36], which we also took into account in our scoring system.

Our results are contrary to the findings of J. Sugar, H.E. Kaufman and E.D. Varnell [37], where they reported that human leukocyte INF was ineffective in either challenge or recurrence HSV studies in rabbits. This can be explained by the difference in the species of IFNs, as well as the difference in the strains of the virus used. In contrast, in the owl monkey eye, the exogenous administration of human leukocyte IFN was clearly effective in preventing HSV-1 infection of the cornea.

Our topical ocular antiviral formulation Oftalmoferon^®^ forte has not exhibited greater therapeutic effects than ACV. ACV and its analogues are the current drugs of choice. Oftalmoferon^®^ forte with different mechanism of action may be good addition for combined drug therapy to minimize the risk of development of the resistance to ACV. In the future, combination studies for in vivo antiviral efficacy should be performed to optimize regimes of treatment of ocular HSV infection.

Finally, our in vivo results show excellent antiviral efficacy of Interferon Vaginal Tablets containing interferon alpha 2-b in controlling HSV-2 using a murine model of vaginal infection. This treatment reduced death of animals, weight loss and clinical signs and pathological manifestations of infections induced by HSV-2. It is interesting to note that the efficacy of Interferon Vaginal Tablets was dose-dependent and was more pronounced when used at a concentration of 6,250,000 IU/kg, 4,166,667 IU/kg and 3,125,000 IU/kg through the intravaginal route. In our study, we used a mouse model, which is widely used to simulate genital infection caused by HSV-2, due to the presence of the HVEM receptor (herpes virus entry mediator) on the cell surface, which is also characteristic of humans [38,39]. The manifestation of herpes infection of the genital tract in humans and mice is similar, but there are some differences. The herpetic ulcers at the vaginal mucosa are typical for humans [14,17], but such sign is not observed in mice. However, the mouse model has more pronounced neurological pathology caused by ascending herpetic lesion of the nerve plexuses and, subsequently, the central nervous system, leading to the death of the animal [40,41]. This fact we observed in our experiments and was taken into account in the scoring system, which we used to estimate the vaginal HSV-2 infection.

It should be noted that the tablet dosages against HSV-2 genital infection in mice were above the value of CC_50_ that we obtained in Vero cells. It is important that we did not observe any toxic effects in mice. This can be explained by several reasons, one of which is the difference between Vero cells and tissues that form the vaginal tract.

In conclusion, our study establishes the antiviral activity of novel IFN formulations against HSV infections in Vero cells. Oftalmoferon^®^ forte shows promise in treating HSV-1 keratitis, while Interferon Vaginal Tablets exhibit significant activity against HSV-2 genital herpes. These topical treatments offer potential advantages in managing HSV infections, especially in cases resistant to conventional therapies. Further research into combination therapies and optimization of treatment regimens is warranted to enhance the clinical utility of these formulations.

## Figures and Tables

**Figure 1 viruses-16-00989-f001:**
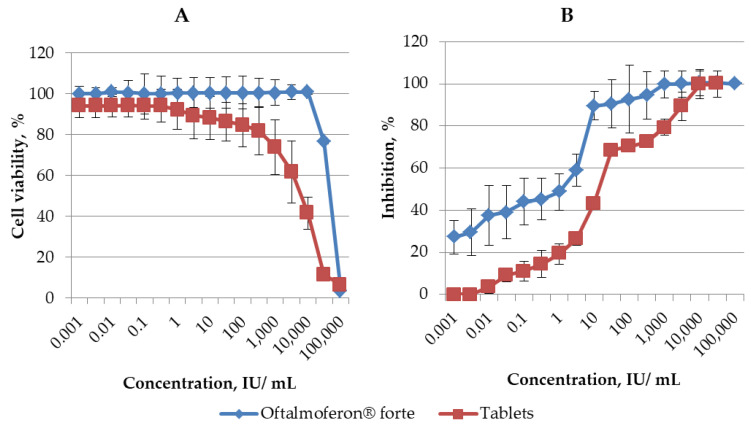
(**A**) Cytotoxicity of Oftalmoferon^®^ forte and Interferon Vaginal Tablets in Vero CCL 81. Cell lines were seeded into 96-well microtiter plates and incubated until formation of a confluent monolayer. Oftalmoferon^®^ forte and Interferon Vaginal Tablets were added at various concentrations. The impact of treatment on cell viability was assessed by MTT assay after 72 h incubation. CC_50_ values were generated and represented as mean ± SD. (**B**) Antiviral activity of Oftalmoferon^®^ forte against HSV-1 and Interferon Vaginal Tablets against HSV-2. Oftalmoferon^®^ forte and Interferon Vaginal Tablets were added to Vero CCL 81 cells at various concentrations. After 2 h of incubation, viruses were added to wells, except “cell control” wells. Cells were incubated at 37 °C in a humidified 5% CO_2_ atmosphere before the appearance of a clear CPE in viral control cells (3–4 days). The CPE and rate of cell viability were measured using MTT assay. EC_50_ values were generated and represented as mean ± SD.

**Figure 2 viruses-16-00989-f002:**
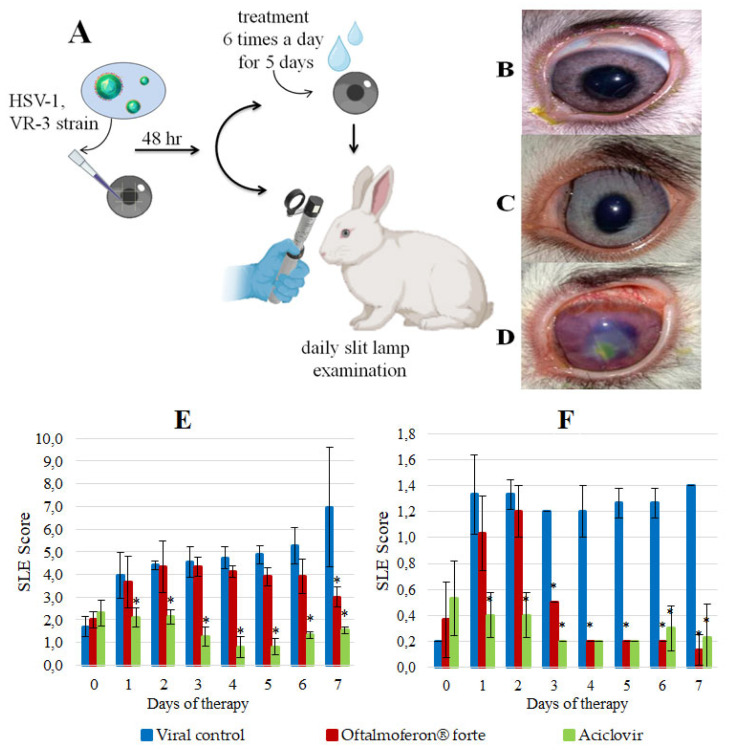
In vivo efficacy of Oftalmoferon^®^ forte (OFT) as a treatment for HSV-1 infection. (**A**) Study design. Rabbits, 2–3 months old, were infected with HSV-1 in the presence or absence of OFT. Treatment began on post-infection (PI) day 2 and continued for five consecutive days. Treatment was applied topically seven times per day. Slit lamp examination (SLE) was performed daily; (**B**–**D**) Rabbit eyes on day 5 of therapy: (**B**) Rabbit from the ACV control group, (**C**) Rabbit from the OFT treatment group, (**D**) Rabbit from the virus control group; (**E**) SLE scores in a rabbit model of HSV-1 stromal keratitis. Asterisks denote significant difference by nonparametric Kruskal–Wallis test with Bonferroni correction * *p* ≤ 0.05; (**F**) SLE scores in a rabbit model of HSV-1 epithelial keratitis. Asterisks denote significant difference by nonparametric Kruskal–Wallis test with Bonferroni correction * *p* ≤ 0.05.

**Figure 3 viruses-16-00989-f003:**
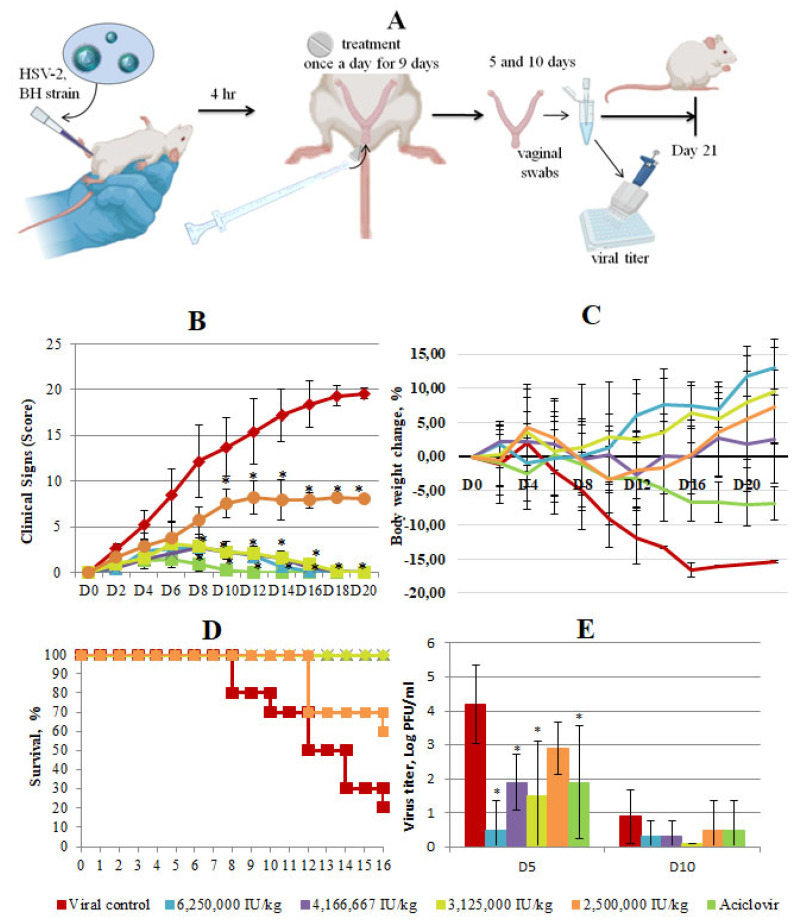
In vivo efficacy of Interferon Vaginal Tablets as a treatment for HSV-2 infection. (**A**) Study design. Female 6-week-old BALB/c mice were infected with HSV-2 in the presence or absence of Interferon Vaginal Tablets. Vaginal tablets were administered intravaginally. At 5 and 10 days post-infection (dpi), mice genitals were swabbed to detect viral titer. The duration of the experiment was 21 days. (**B**) Clinical scoring of animals was performed daily on a score system until the end of the experiment. Scores were assigned using the following scoring system: hyperemia 0–3, edema 0–3, discharge 0–3, lesions of the peripheral and central nervous system 0–20. Each score was added to provide a total score for each specific day. Data represent means ± SD. Asterisks denote significant difference by nonparametric Kruskal–Wallis test with Bonferroni correction * *p* ≤ 0.05. (**C**) Weight change in mice infected with 10^6^ TCID_50_/mL of HSV-2 in percentage, normalized to the body weight at the time of infection. There was high variability and as such error bars are not displayed to improve clarity, (**D**) Kaplan–Meier survival plot of all mice groups at different days after HSV-2 infection, (**E**) Secreted virus titers assessed from the swabs of vaginas 5 and 10 days post-infection. Asterisks denote significant difference by non-parametric Kruskal–Wallis test with Bonferroni correction * *p* ≤ 0.05.

**Table 1 viruses-16-00989-t001:** Antiviral activity of Oftalmoferon^®^ forte and Interferon Vaginal Tablets against of HSV-1 and HSV-2, respectively.

Formulations	Viruses	CC_50_, IU/mL	EC_50_, IU/mL	EC_90_, IU/mL
Oftalmoferon^®^ forte	HSV-1	56,647.08	4.24	38.16 ± 1.58
Interferon Vaginal Tablets	HSV-2	5437.53 ± 530.74	6.98 ± 2.39	62.82 ± 1.80

**Table 2 viruses-16-00989-t002:** SLE scoring system for stromal keratitis caused by HSV-1 in rabbit eyes.

Characteristics	SLE Score
Conjunctival edema	Normal—0; mild—0.5; moderate—0.7; significant −1
Corneal edema	Normal—0; local—0.1; perelimbal—0.2
Edema of the iris	Normal—0; mild—0.1; moderate—0.2; significant −0.3
Conjunctival hyperemia	Normal—0; mild—0.5; moderate—0.7; significant −1
Hyperemia of the iris	Normal—0; mild—0.1; moderate—0.2; significant −0.3
Vascularization	Normal—0; surface—0.3; deep—0.5; mixed—0.7
Corneal roughness	Normal—0; mild—0.1; moderate—0.3; significant—0.5
Discharges from the eyes	Normal—0; catarrhal—0.1; fibrinous, small volume—0.3; fibrinous, significant volume—0.4; purulent–fibrinous, small volume—0.5; purulent–fibrinous, moderate volume—0.7; purulent–fibrinous, significant volume—1

**Table 3 viruses-16-00989-t003:** SLE scoring system for ulceration caused by HSV-1 in rabbit eyes.

SLE Score	Characteristics of Ulceration
0.2	punctate, isolation, small size, in places of scarification
0.5	punctate, middle size, in places of scarification
0.7	punctate, isolation, small size, in places and beyond of scarification
1	punctate, middle size, in places and beyond of scarification
1.2	converging ulcerations, middle size, in places and beyond of scarification
1.4	punctate and converging ulcerations involved 1/4 of the cornea
1.6	punctate and converging ulcerations involved 1/3 of the cornea
1.8	punctate and converging ulcerations involved 1/2 of the cornea
2.0	punctate and converging ulcerations involved all of the cornea

**Table 4 viruses-16-00989-t004:** Clinical symptoms of genital herpes in mice.

Clinical Symptoms	Score
Hyperemia	Normal—0; mild—1; moderate—2; significant -3
Edema	Normal—0; mild—1; moderate—2; significant -3
Discharge	Clear, watery or mucous—0; Abundant, clear, watery or mucous—1; Clear, thick, gel-like inclusion -2 Flaky whitish discharge—3
Lesions of the peripheral and central nervous system	Normal muscle strength—0, The early stage weak muscle tone reduction with full movement—1; The second stage, significant reduction in movements in the joint, muscles are able to overcome gravity and friction (the possibility of lifting the limb from the surface)—2; The third stage, movement in the joint is significantly reduced, movement without lifting the limb, crawling—3; The fourth stage, barely noticeable muscle contractions, no movement in the joints—4; The fifth stage, no voluntary movements, monoplegia (paralysis of the first limb)—5; Paraplegia (paralysis of both limbs)—10; Death—20.

## Data Availability

The data presented in this study are available upon demand from the corresponding author.

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
