# Peer review of "The Topical Novel Formulations of Interferon α-2в Effectively Inhibit HSV-1 Keratitis in the Rabbit Eye Model and HSV-2 Genital Herpes in Mice"

_viruses, 2024, doi:10.3390/v16060989_

Round 1
Reviewer 1 Report
Comments and Suggestions for Authors
Please ss the attachment

Author Response
- In Figure 1 legend, the Fig.1B is missing. The inhibition against viral infection should be presented directly. 100% inhibition dosage should be identified. The standard plaque assay should be used to examine the antiviral activity of these formulation.
Thanks for the comment. We apologize for inaccuracy in this part. The caption 1B was added to Figure 1.
The 100% inhibitory effect was reached and we designated it on the curve on the graph (Figure 1B). It is generally accepted that EC 50 is enough to express antiviral activity in vitro, but we have calculated and included EC90 according to your request (Table 1). Previously, we used the plaque assay to determine antiviral activity, but we found a quantitative MTT assay, that estimates inhibition of the cytopathic effect more reliable and convenient. Earlier we compared both assays in another experiments and found a good correlation between them. MTT assay is also widely used for study antiviral activity against HSV in cell culture ( Hitoshi Takeuchi, Masanori Baba, Shiro Shigeta, An application of tetrazolium (MTT) colorimetric assay for the screening of anti-herpes simplex virus compounds, Journal of Virological Methods, Volume 33, 1991, Pages 61-71; Silva-Mares D, Rivas-Galindo VM, Salazar-Aranda R, Pérez-Lopez LA, Waksman De Torres N, Pérez-Meseguer J, Torres-Lopez E. Screening of north-east Mexico medicinal plants with activities against herpes simplex virus and human cancer cell line. Nat Prod Res. 2019 May;33(10):1531-1534. doi: 10.1080/14786419.2017.1423300. Epub 2018 Jan 15.; Müller V, Chávez JH, Reginatto FH, Zucolotto SM, Niero R, Navarro D, Yunes RA, Schenkel EP, Barardi CR, Zanetti CR, Simões CM. Evaluation of antiviral activity of South American plant extracts against herpes simplex virus type 1 and rabies virus. Phytother Res. 2007 Oct;21(10):970-4. doi: 10.1002/ptr.2198.)
- In Fig. 1A, cell viability should be [OD of treated cell/OD of untreated cell] x100. If the cell viability is calculated as stated on page 3, lines 110-111 (see below), a concentration of 0.001 to 0.1 will have a similar OD as the untreated cells. The final cell viability will be close to 0.
cell viability (%) = [(OD of untreated cells - treated cells)/(OD of untreated cells)] × 100.
For determination of cytotoxicity of drugs we used the formula Cell viability (%) = [(OD of untreated cells - treated cells)/(OD of untreated cells)] × 100. In this formula, it is not the percentage of inhibition that is considered, but the percentage of cell viability. Our cell viability rate for concentration of 0.001 to 0.1 is close to 100% similar to untreated cell control (1 A cytotoxicity of drugs). With increasing concentration, cell survival decreases and at high concentrations it is close to 0 that means that almost all cells are dead and this concentration of drug is toxic for cells. For example, according to our formula, if the inhibition is 80% then:
cell viability (%) = [(OD of untreated cells 1,0 - treated cells 0,2)/(OD of untreated cells 1,0)] × 100 = 80%
- In Table 1, it should have a note for CC50 and EC50, respectively. The CC50 and EC50 were only based on 10 TCID50 for HSV-1 and 100 TCID50 for HSV-2. How about a higher dosage of HSV-1 and HSV-2? How about 5X105 TCID50/eye of HSV-1 or 2X105 TCID50/ml of HSV-2?
In our research, we used two different viruses that develop cytopathic effects in cell culture in different ways. For each of these viruses, such multiplicities of infection were chosen that give a pronounced cytopathic effect in cell culture and are well seen in the quantitative MTT test. In the case where we used higher doses of infection, cytopathic effect developed too quickly and was not available for quantification, and in the case of lower doses of the virus, cytopathic effect developed too slowly and was not detectable. Besides in our study we used Vero cells that differ from the tissues forming the eye, so we did not transfer the dose of infection from the cell culture to the animal model. In the animal model, we used viruses with a titer of 106TCID50/ml. Increasing the doses of infection would make sense if we used a cell culture similar to the cells of the eye or vaginal tract.
- In Tables 2 and 3, SLE scoring systems should include images for the scoring scales. How about dendritic lesions in rabbits infected with HSV-1?
Thank you for your offer. During the work, we did not take images of the lesions the difficulty of taking photos through a slit lamp. When providing the material, we referred to an paper in which such data was also presented as Table. (Banmeet S. Anand, James M. Hill, Surajit Dey, Koichi Maruyama, Partha S. Bhattacharjee, Marvin E. Myles, Yasser E. Nashed, Ashim K. Mitra; In Vivo Antiviral Efficacy of a Dipeptide Acyclovir Prodrug, Val-Val-Acyclovir, against HSV-1 Epithelial and Stromal Keratitis in the Rabbit Eye Model. Invest. Ophthalmol. Vis. Sci. 2003;44(6):2529-2534. https://doi.org/10.1167/iovs.02-1251 )
Pronounced dendric lesions of the cornea are more typical for humans, dendric lesions in rabbit eyes depend on the degree of damage, the rate of healing of the epithelium of the eye and age of animal. The old rabbits have slow process of regeneration of the corneal epithelium and dendritic lesions are well seen using a slit lamp. In our experiment, we used young rabbits 2.5-3 months old, whose corneal regeneration rate is very high and we did not observe clearly dendric lesions using slit lamp. The fine subtle small corneal defects healed very quickly, since the erosions and ulcers were well seen. This is why we concentrated on these signs of infection.
- In Table 4. lesions of the peripheral and central nervous system at different stage should be presented as images, either gross images or histopathology images.
We did not take images of nerve lesions in mice. We presented the data as Tables, following the example of the article Banmeet S. Anand, James M. Hill, Surajit Dey, Koichi Maruyama, Partha S. Bhattacharjee, Marvin E. Myles, Yasser E. Nashed, Ashim K. Mitra; In Vivo Antiviral Efficacy of a Dipeptide Acyclovir Prodrug, Val-Val-Acyclovir, against HSV-1 Epithelial and Stromal Keratitis in the Rabbit Eye Model. Invest. Ophthalmol. Vis. Sci. 2003;44(6):2529-2534. https://doi.org/10.1167/iovs.02-1251.
- In Figure 3, how the Interferon Vaginal Tablets were administered is not clearly described. The effective tablet dosages against HSV-2 genital infection in mice are above the CC50 tested in Fig. 1. Would that be a concern? It should be discussed in the end.
Thanks for the guidance. We have changed the image for greater clarity of understanding, and also added the description to the text of Chart of Figure 3 (Vaginal tablets were administered intravaginally). Cytotoxicity was determined in Vero cell culture for further studies of antiviral activity in vitro. In this case, we used the indicated dosages on an animal model of mice and no toxic effects were found in animals. Anyway we thank the reviewer for the very interesting suggestion and added some comments concerning it in Discussion
Reviewer 2 Report
Comments and Suggestions for Authors
The Authors in this report describes a drug containing Interferon α-2β which is used as a topical treatment for HSV-1 in infected rabbits and Interferon vaginal tablets for treating HSV-2. The study in combination targets both HSV-1&2 which demands serious attention in current era but the methods used in this study does not effectively defines the target. Concerns to be addressed:
1. The drug applied to infected rabbits started a day before infection. It then continued for 5 consecutive days atleast six times a day. This seems a lot of drug in the eye and at the same time not realistic to treat clinical symptoms.
2. The drug was used prophylactic and therapeutically as well which seems like a forced measure to show some alleviation. Not sure, how can it treat patients in need.
3. Line 137 is confusing, females of what age?
4. Did the authors study which pathway does the drug target?
5. Authors talk about antiviral activity, Acyclovir already targets that, therefore, I don't see a novelty in that.
6. Line 64, please correct for grammar. Spellings of Acyclovir are wrong in multiple places.
7. Line 286, please correct the word treated.
Comments on the Quality of English Language
requires major editing.
Author Response
- The drug applied to infected rabbits started a day before infection. It then continued for 5 consecutive days atleast six times a day. This seems a lot of drug in the eye and at the same time not realistic to treat clinical symptoms.
Thanks for the comments. When we chose a treatment regimen, we were guided by the schemes used with other formulations of Interferons licensed in Russia. In clinical cases, it is recommended to administer the drug 6 times a day, therefore, causing an acute infection, we used the same treatment regimen. In addition, for the topical formulation of Acyclovir, the recommended frequency of administration is 5 times a day in clinical practice.
- The drug was used prophylactic and therapeutically as well which seems like a forced measure to show some alleviation. Not sure, how can it treat patients in need.
In our work, the study of efficacy of drugs was performed experimentally in an animal model and further clinical trials will show the final result.
- Line 137 is confusing, females of what age?
Thanks for the comments. We have corrected the sentence in the text: Female rabbits of the Soviet chinchilla, 2.5-3 months old
- Did the authors study which pathway does the drug target?
We did not study the pathways that our formulations target. It was outside the scope of our study. The main component of studied formulations is Interferon α-2в, the mechanism of action of Interferon is known. Interferon exhibits antiviral activity, launching a program of synthesis of antiviral proteins in cells and exhibits immunomodulatory properties. Type I interferon is signaled through its dimeric interferon α/β receptor (IFNAR) to induce the JAK/STAT signaling cascade, which includes transcription of interferon-stimulated genes (ISG) to inhibit viral replication.And type I interferon indirectly recruits inflammatory monocytes during HSV-1 and HSV-2 infection, contributing to survival and antiviral response, which was shown in a mouse model (Gill N, Deacon PM, Lichty B, Mossman KL, Ashkar AA. Induction of innate immunity against herpes simplex virus type 2 infection via local delivery of Toll-like receptor ligands correlates with beta interferon production. J Virol. 2006;80(20):9943-9950. doi:10.1128/JVI.01036-06 , Conrady CD, Halford WP, Carr DJ. Loss of the type I interferon pathway increases vulnerability of mice to genital herpes simplex virus 2 infection. J Virol. 2011;85(4):1625-1633. doi:10.1128/JVI.01715-10).
- Authors talk about antiviral activity, Acyclovir already targets that, therefore, I don't see a novelty in that.
We appreciate the reviewer’s comment and agree that Acyclovir is gold standard in HSV therapy. However the emergence of resistance to Acyclovir, particularly in immunocompromised individuals, poses a significant challenge. The mechanism of action of Acyclovir differs from the mechanism of action of interferon. Acyclovir is a direct antiviral drug that blocks the synthesis of viral DNA. Interferon is an indirect antiviral drug that causes suppression of virus replication in infected cells, inhibition of cell proliferation, and also has an immunomodulatory effect. So topical formulations of Interferon may be useful as a maintenance and replacement therapy. In addition, the combination therapy Acyclovir and Interferon may be promising. We have noted it in Discussion
- Line 64, please correct for grammar. Spellings of Acyclovir are wrong in multiple places.
We regret the errors and have made the correction throughout the text.
- Line 286, please correct the word treated.
We have made the correction.
Round 2
Reviewer 1 Report
Comments and Suggestions for Authors
In this revised article entitled: The topical novel formulations of Interferon α-2в effectively inhibit HSV-1 Keratitis in the Rabbit Eye Model and HSV-2 Genital Herpes in mice, Ivanina A. et al. have made some revisions to the reviewer’s critiques. Most of the concerns were discussed in the cover letter.
Overall, this revised article did not provide extra solid data to support the antiviral effect of the Interferon α-2в. The effect of Oftalmoferon® forte against virus shedding was not examined in the Rabbit eye model. The EC50 and EC90 were established only against HSV-1 and HSV-2 at 10 and 102 TCID50/ 100µL, respectively. The EC50 or EC90 will be different when the virus concentration increases.
The following are specific issues to be addressed.
- Acyclovir was named as Aciclovir in both Fig. 2 and Fig. 3.
- Line 109-111: Cell viability (%) = [(OD of untreated cells - treated cells)/(OD of untreated cells)] × 100. This Formula still does not make sense for Fig. 1A. Cells treated with lower concentration will have little cytotoxicity and will have OD similar to that of untreated cells. The top numerator will be 0. The final cell viability will be 0 with the use of your Formula. It should be (OD of treated cell/OD of untreated Cell) X100
- Line 148: (2 caplets 50 µL at concentration of 1%). What does the sentence mean within the parenthesis?
- Lines 297-300: However, the complete disappearance of clinical symptoms in the group treated with ACV 298 occurred earlier (on day 12 after infection compared with on day 16-18 after infection in 299 groups treated with Interferon Vaginal Tablets (Fig. 3 B). Where does the parenthesis end?
Author Response
Response to reviewer
- Acyclovir was named as Aciclovir in both Fig. 2 and Fig. 3.
We regret the errors and have made the correction.
- Line 109-111: Cell viability (%) = [(OD of untreated cells - treated cells)/(OD of untreated cells)] × 100. This Formula still does not make sense for Fig. 1A. Cells treated with lower concentration will have little cytotoxicity and will have OD similar to that of untreated cells. The top numerator will be 0. The final cell viability will be 0 with the use of your Formula. It should be (OD of treated cell/OD of untreated Cell) X100
We greatly appreciate the thoughtful comment. You are absolutely right, our Formula that we pointed in Methods is not correct. We have not change the Figure because all our calculations were made using your right formula. We regret about misunderstanding in your first time review.
- Line 148: (2 caplets 50 µL at concentration of 1%). What does the sentence mean within the parenthesis?
Thank you for your comments. We clarified the doses that we used in our study.
- Lines 297-300: However, the complete disappearance of clinical symptoms in the group treated with ACV 298 occurred earlier (on day 12 after infection compared with on day 16-18 after infection in 299 groups treated with Interferon Vaginal Tablets (Fig. 3 B). Where does the parenthesis end?
Thank you for your comments, it was a mistake and we removed parenthesis
Reviewer 2 Report
Comments and Suggestions for Authors
A comparison report should be cited where the efficacy of the drug has been measured elsewhere.
Comments on the Quality of English Language
fine.
Author Response
Thank you for your comments!